# Two angles of overqualification-the deviant behavior and creative performance: The role of career and survival job

Nasib Dar, Wali Rahman *

Department of Business Administration, Sarhad University of Science & Information Technology, Peshawar, Pakistan

* wali.ba@suit.edu.pk, mayarwali@gmail.com

## Abstract

Overqualification has both positive and negative outcomes; however, extant literature exhibits a tilt in favor of its negative aspect against its positive side. This tilted approach results in derision of a condition which could produce positive results. We argue, through empirical evidence, that there might be some circumstances/conditions, like the intentions of employees about their current job, that may play an important role in enabling organizations to be benefitted from the surplus KSAs of the overqualified employees and overqualification can be used as a tool to mitigate the effects of its negative consequences. To empirically test this condition, a sample of 193 teachers and 193 students have been selected through cluster sampling technique. The results exhibited that if overqualified employees perceive their current job as a career job then there is a more likelihood that they will be involved in some innovative and creative behaviors instead of deviating negatively from the organization norms. The results provide some new research insights that could be used to better understand the phenomenon of overqualification. The results of the study have practical implications for HR managers.

## Introduction

Over the few decades the saturated labor market and fierce competition, complementing each other, have forced the job seekers to get highest qualification to ensure entry as well as survival in the market. The result is labor market has, mostly, become overqualified [1].This means most of the job seekers have surplus knowledge, skills and abilities (KSAs). Formally Perceived overqualification refers to the perception of having excess knowledge, skills and abilities than required by one's current job [2]. This phenomenon is more evident in countries where economies are in poor conditions [2–5] and issues like age security, high unemployment, absence of unemployment compensation [1] leave individuals with no option except to get job that are below their working capabilities. This conditionof having over KSAs is generally considered problematic because empirical evidences bespeak that such employees are less committed and have been found less satisfied [e.g., 6, 7, 8]. It has also been found that these employees are more prone to leave their organizations voluntarily [3, 8–10]. And because of these problems,

**Data Availability Statement:** All data files have been uploaded into the system as Supporting Information files.

**Competing interests:** The authors have declared that no competing interests exist.

overqualified individuals happen to be undesirable workers [11] as they join misfit job by chance not by choice.

This mis-fitness, as evident from empirical researches, has been reported as having many negative outcomes such as negative job attitudes [12, 13] destructive deviance [14], voluntary turnover [3, 8], and actual turnover [1, 3, 12]. Notwithstanding, these findings, in the unprecedented competitive global markets, could hardly deter organizations in their search for highly qualified and experienced employees. Here the driving force is the set of expectations attached with extra KSAs like, creativeness and other constructive outcomes. And there are glimpses [e.g., 12] in literature wherein creative performance has been shown as the most common positive outcome of overqualification. So, this paradoxical situation invites for exploring answer to the questions as why, when and how overqualified employees will be destructive and/or constructive.

No doubt some researchers [e.g., 12] have made a few attempts to explore its positive outcomes like being creative. However, empirical evidence of its link with positive outcomes is still lacking and the area is under-explored [15]. These mix findings portend that overqualification need not be looked upon, always, with negativity. Research, wherein it is seen with a positive lens, appears on the rise [16]. Researchers [e.g., 17, 18] contend that through some mechanisms and/or under some conditions, overqualification is linked with high performance. This means that surplus KSAs of the overqualified employees can benefit organizations. Therefore, keeping in view both angles of overqualification, different researchers [e.g., 2, 19] call for introducing some variables as moderators in order to better understand the nature of relationship of overqualification with its outcomes. Thus in line with these calls, this study takes into consideration intentions of employees towards the job (either it is joined as career or survival job) as moderator between perceived overqualification as independent variable with deviant work behaviour and perceived creative performance as dependent variables.

This research adds to the literature of overqualification in three specific ways. First, it addresses the noticeable gap in literature of overqualification and its relationship with its positive outcomes [2]. No doubt, there are some studies that have looked into some personal factors like marital status [20], gender and self-esteem [21], some contextual moderators such as empowerment [2] and emotional support [22] that affect the relationship of overqualification with its outcomes like job satisfaction, intention to remain, voluntary turnover and future self-esteem. But no attempt has been made that what role the intentions of overqualified employees regarding current job might play. Therefore this research takes career stakes (i.e., career or survival job) as moderator to fill this gap and made a minute addition in the literature of overqualification. The career stakes refers to "the degree of commitment to one's current job as a long-term employment trajectory [23]. In case of career job the employment relationship bound with personnel commitment along with structure commitment i.e., binding workers to jobs [24, 25]. The career stakes affect the deviant and conforming actions of employees at workplace [23].

Second, to-date the performance of overqualified employees has been reported and measured only either by their self-reports or by supervisor reports [12, 26]. But this research attempts to measure the creative performance in terms of how the customers rate overqualified individuals in terms of their creativity. Such measurement will be more objective in the sense that the response for measuring performance will be taken from the customer directly in order to avoid the bias of subjective rating which might be more in case of self-rating or supervisor rating.

Finally, the study is conducted in Pakistan. Majority of studies on overqualification has been conducted in West [27] and very few are outside of the West [2]. However, the issue of overqualification is not limited only to industrialized countries but developing countries are

also under its influence. The population for this research is employees of elementary and secondary education department of KP Pakistan. The reason of selecting sample from this sector is that in previous five years (2013–2018) the KP Government has hired, nearly, fifty thousand teaching staff. The required qualification for primary level position was BA/BSc. (bachelor level education) while for high level it was master degree. However, majority of the selected candidates have higher academic qualification (having M Phil, MS and PhD degrees) and are still working. This means, so for, no voluntary withdrawal has been observed. To consider this context the results might be more objective and generalizable.

## Theoretical background

Majority of the researchers, who have studied overqualification with its negative consequences and how these could be mitigated, have grounded their respective research taking into consideration the two theories—equity theory [28] and relative deprivation theory [29]. No doubt, these theories provide some explanation for employee behaviour when overqualification is instrumental in engendering negative behaviour; these theories are inadequate to explain as to when and how overqualification yields positive consequences. Therefore, researchers [16] contend that there is a need of a theoretical perspective which may cover the positive consequences of overqualification as well. So, to have key insight on the potential positive consequences of overqualification, there is a need of taking into consideration some additional theories like social identity theory [30]; human capital theory [31]; and social learning theory [32]. Taking these theories into consideration will help to illuminate the brighter side to the story [16].

In addition to the above theories, the current research posit that resources drain theory [33, 34] is also relevant to the issue of overqualification because an overqualified employee tends to move resources from one domain to other domains which include personal pursuits [33]. It is, generally, believed that in order to fully satisfy their needs of challenging and intrinsically motivating jobs, overqualified employees might be in continuous search for jobs that best fit to their KSAs. Having this intentions in mind and active behavior for search of fit-job they are unable to fully concentrate on current job and do not utilize time and energy in current job and are more likely involved in deviant (negative) behaviors such as coming late, withholding efforts, absenteeism etc. So the major driver of deviance here is not the stress but their intentions for satisfying their desired needs of getting fit job. We posit that this often happens when job is taken as a survival job. However, we argue that if this relationship is moderated by job as career job, then it may drive them towards positive consequences (creative performance). Based on the passed tested theories and this additional dimension of resources drain theory, the current research sets a number of hypotheses and then empirically tests the same to answer the how and when questions related to positive consequences of overqualification.

## Literature review

### Deviant behavior and overqualification

In literature workplace deviance has various definitions with minute differences. We would pick one [35] because these definitions have conceptual common threads. In the words of Robinson and Bennet [35] workplace deviance is a voluntary behavior that violates significant organizational norms and in doing so the well-being of an organization, its members, or both are threatened. However, it seems hasty conclusion to say that voluntary deviation from organizational norms would always end in harming the well being of the stakeholders. This deviation could be creative and innovative. Humans need not be robots. Creativity and innovativeness are two distinct positive qualities [36]. The former refers to creating something new

(i.e., generation of new ideas) while the latter is concerned with the implementation of the new stuff (i.e., implementation of new ideas) in work setting. Consistent with this conceptualization in current study the creative performance is defined as the generation of "products, ideas, or procedures that satisfy two conditions, namely, they are novel or original, and they are potentially relevant for, or useful to, an organization" [37]. This study aims to explore under what boundary conditions the overqualified employees would deviate constructively or destructively.

Similarly, Overqualification is a quality of an employee wherein "the individual has surplus skills, knowledge, abilities, education, experience, and other qualifications that are not required by or utilized on the job" [19]. The literature on overqualification has very mix findings of its being a positive or a negative phenomenon [38]. Using different mechanisms (e.g., relative deprivation theory, equity theory and person-job-fit] several studies have found that overqualified individuals feel frustration; stress and anger in work setting which further results into various psychological consequences [22]. For example, taking person job-fit as theoretical foundation, researchers [14] have empirically found positive relationship between perceived overqualification and counterproductive work behavior. They illustrated that overqualified employees perform their core duties well but have no space and opportunity for utilizing their valued skills and to satisfy their need of challenging and intrinsically motivating jobs, such conditions lead to strain and harmful behaviors. Later studies [1] have complemented such findings with somewhat similar results of positive relationship between perceived overqualification and counterproductive work behaviour. This suggests that overqualified employees, having stress, when try to overcome the stress get involved in some sort of harmful behaviors. Having this intention in mind and active behavior for search of fit-job they are unable to fully concentrate on current job and are more likely get involved in deviant behaviors such as coming late, withholding efforts, absenteeism etc. Leaving the contention of whether overqualification is constructive or destructive under any boundary conditions, we would put the following hypothesis to test to empirically validate the positive relation between the two:

**H1**: *Perceived overqualification will have a positive relationship with deviant behavior.*

## Overqualification and creative performance

The literature is replete with researches that exhibit the relationship of overqualification with negative consequences. Nevertheless, researches have also empirically tested its relationship with positive outcomes like its positive impact on performance. It is contended that overqualified employee will perform well on task level because task performance has relevance with KSAs but it is conditioned with their motivation [39]. Researchers [e.g., 38, 40] have found that overqualification has a positive relationship with creative performance. These researchers argued that while having enough KSAs overqualified employees would easily perform their core activities and, therefore, have enough time and opportunities to use their valued skills in creative process require for creative performance [15]. However, creative performance is conditioned to some contextual factors like POS, opportunities to mentoring others and development of new ideas. The findings of researchers [15] are worth noticing in the sense that they are taking into consideration the context where an employee is working. It is a signal towards this rational that overqualified employees would perform constructively if they are provided with the suitable environment. However, we nuance that it is not only the context that has value, it is the intentions of employees that also matter. Based on the extant literature, this study would put the following hypothesis for empirical testing and validation of the previous findings:

**H2**: *Overqualification has a positive relationship with creative performance.*

## The condition of perception of job type (career or survival job)

Looking into the relationship of overqualification and its positive or negative consequences, researchers have also explored some conditions (the moderating effects of moderators) that could affect this relationship. Such studies are numerous in terms of numbers and very informative in terms of quality [19]. Researchers [2, 19] contend that by inserting moderators (e.g., negative job attitudes and behaviors) between overqualification and its outcomes, it would be possible to mitigate its negative effects and could boost its positive results. To this effect various studies have used different moderators (conditions) between overqualification and its outcomes. For instance, Johnson and Johnson [22] have used supportive environment; Erdogan Kraimer and Liden [41] have employed quality of relationship between a leader and subordinate; and Erdogan and Bauer [2] have used psychological empowerment as moderators which mitigate the negative effects of overqualification. These are all conditions. In the same line, we opine that employee's perceptions of job type (career or survival job) are two conditions and can affect the relationship as moderators. These two conditions can have their corresponding affects i.e., the creative performance (positive effect) and deviant behavior (negative effect) respectively.To elaborate it, we argue that if an overqualified individual joins a job as a survival job, in such condition the employee will be more likely involved in deviant behaviors. It is because according to resources drain theory [33] an overqualified employee will be in continuous search of career job and will not be able to invest his/her full time and energy in the current job which ultimately results in deviant behaviors. On the other hand, researchers [23] assert that when employees join a job as a "career job" and not as a "survival job" are less likely involved in deviant behaviors. Taking this as a base, we argue that if an overqualified individual joins a job as a career job will be more likely involved in creative process, because they will be able to perform their core work with less time and energy and will have enough time and energy for creative process. Ultimately, they will perform creatively. Therefore, the H1 and H2 are conditioned based hypotheses with career and survival stakes and further this study proposes the following hypotheses.

**H3a**: *The perception of job as career job negatively moderates the relationship of overqualification and deviant behavior.*

**H3b**: *The perception of job as survival job positively moderates the relationship of overqualification and deviant behavior.*

**H4a**: *The perception of job as career job positively moderates the relationship of overqualification and creative performance.*

**H4b**: *The perception of job as survival job negatively moderates the relationship of overqualification and creative performance.*

For sake of ease, all the hypotheses are, collectively, modeled in Fig 1.

## Methods

### Setting, participant and procedures

To test the hypotheses, education department of Khyber Pakhtunkhwa, (a province of Pakistan) has been taken as population of the study. In this province the provincial government has hired nearly fifty thousand new teaching staff in the previous five years (2013–2018). Teachers have been hired in two categories—primary lever and secondary level—for which B. A./BSc. (14 years education) and Master (16 years education) respectively were required as the base qualification. However, actually a considerable number of the hired staff has Master degree, MS/M.Phil and PhD degrees in the respective levels. This means this class is overqualified. The reason of this overqualification is that universities in the province have been producing thousands of undergraduates, graduates and postgraduates for the already saturated job

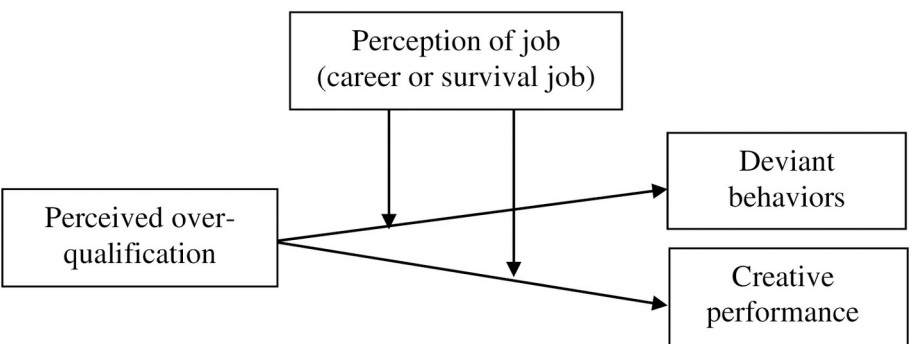

**Fig 1. Hypothesized model of the study.** This figure pictorially presents the relationships of the variables of the study.

market; consequently candidates join these jobs just for survival and due to the fear of over aging. Notwithstanding, all employees are independent to work creatively and there is no compulsion on them for doing so. Thus the context of the selected sample is suitable to measure the study variables. To study the population, cluster (district) sampling technique was used. Within the cluster to choose schools purposive sampling technique was employed.

Instead of studying the whole population, only one district (District Dir) has been taken as a cluster because this is the biggest district in terms of the overqualified hired teachers. It has nearly 8,000 teachers in the subject case. The creative performance of the selected sample was measured by the response of students on a structured interview in their mother language. The items were translated from English to Pashto by using the expertise of two professors of Pashto and two of English. The data was collected in two waves. In the first wave the data was collected from teachers on perceived overqualification, career stakes and deviant workplace behavior. By doing so the overqualified teachers were identified. In the second wave, after a period of a month their respective students were asked about the creative performance of those specific teachers through structured interview.

The survey questionnaires were distributed among 230 teachers in the Primary, Middle, High and Higher Secondary schools in the target population. Questionnaires distribution and collection took almost two months (February-March, 2019). A total of 204 were received back in which 11 were found with missing information and were discarded. Thus the response rate remained 83%.

So far the size of the sample is concerned; there is no consensus among the researchers and statisticians. However, there are factors like number of variables in the study, size of the population, etc., that affect the determination of sample size. For Sekaran [42] "sample sizes larger than 30 and less than 500 are appropriate for most research. . . . . .in multivariate research (including multiple regression analyses), the sample size should be several times (preferably 10 times or more) as large as the number of variables in the study" (p. 295). While Reisinger and Mavondo [43] recommend a ratio of five to ten respondents for each estimated parameter. The current study followed them in the determination of sample size.

The sample includes only overqualified employees and the rest of the teachers are excluded. Furthermore, participation is voluntary and at the convenience of the participants. All research ethics like confidentiality, privacy, etc. have been taken care of and the participants have been informed with the help of a covering letter attached to the questionnaire.

## Demographic characteristics of respondents

Out of 193 respondents, 61 (32%) were female and 132 (68%) were male. Due to cultural constraints, collecting data from female respondents is relatively difficult that is why the number

of female respondents is comparatively low. The 81 (42%) respondents were of age of < 30 years, 112 (58%) were of age of 30–50 years and no one has more than 50 years of age. The sample of the study was only those employees who have hired during previous 5 and half years, therefore, major component, in terms of age, was of 30–50 years followed by < 30 years while there was none above 50. On the education side, out of 193 respondents 46 (24%) hold Master degrees, 119 (62%) did MS/M.Phil and 28 (14%) had PhD degrees. The reason of selecting candidates having high degrees is that to ensure that their qualification is above than minimum qualification of a position. In terms of experience, 181 (94%) had less than 5 years experience, and 12 (6%) had 5–10 years with non above 10years experience. Most of the employees had less than 5 years experience because most of the respondents are hired in previous % and half years. In terms of marital status, out of 193 the 154 (80%) respondents were married, 35 (18%) were female and 4 (2%) were divorce. Most of the respondents were married because in the target population people culturally prefer to marry in early ages. In addition, among the student-respondents, 72 (37%) were girls and 121 (63%) were boys. The detail of demographic profile of the respondents is presented in Table 1.

## Measures

*Perceived overqualification*: Perceived overqualification was measured by using Maynard et al. [8] nine-item scale. It measures the perception of employees regarding their surplus KSAs (e.g., "I have more abilities than I need in order to do my job". α = 0.92).

*Deviant behavior*: To measure deviant behavior 19-items scale of Bennett and Robinson [44] was adopted. A five point Likert scale was used where responses range was from 1 = "never" to 5 = "always". Out of 19 items 7 items measured the interpersonal deviance and the

**Table 1. Demographic of characteristics of survey respondents (overqualified staff).**

| Sample characteristics | Frequency (n = 193) | Percent (%) |
|---|---|---|
| **Gender** | | |
| Female | 61 | 32 |
| Male | 132 | 68 |
| **Age** | | |
| < 30 years | 81 | 42 |
| 30–50 years | 112 | 58 |
| 50+ years | 0 | 0 |
| **Education** | | |
| Master | 46 | 24 |
| MS/MPhil | 119 | 62 |
| PhD | 28 | 14 |
| **Experience** | | |
| < 5 years | 181 | 94 |
| 5–10 years | 12 | 6 |
| 10+ years | 0 | 0 |
| **Marital status** | | |
| Single | 35 | 18 |
| Married | 154 | 80 |
| Divorce | 4 | 2 |
| **Students respondents** | | |
| Female | 72 | 37 |
| Male | 121 | 63 |

remaining 12 items measured the organization directed deviance. The Cronbach alpha value for deviant behavior is $\alpha$ = 0.94.

*Creative performance*: Creative performance was measured by using Tierney, Farmer, and Graen [45] five-items scale. The scale was supposed to measure employees' potential of generation new and useful ideas. To measure creative performance we employed a structured interview where we asked questions from the students about the concerned teachers. The Cronbach alpha for creative performance is $\alpha$ = 0.76.

*Career stakes (Career or Survival job)*: We have adopted the mechanism of Huiras et al., [23] to measure the career or survival job. In survey, we asked each respondent "How is your present job related to your long-term career goals?" This variable being having two categories, therefore, was dummy coded with 1 (career job) and 0 (survival job).

## Control variables

The effect of demographic factors such as tenure, education and age were treated as controlled variables in this study. This is because some studies have shown that employees having more time in an organization have been found to be less likely involved in deviant acts as compared to those who are relatively new [46]. Moreover, a more educated a person is expected to be involved less in deviant acts [46]. Finally, the age of employees also matters because it has been found that the young employees tend to be more involved in deviant acts like theft as compared to old employees [47]. Gender was also marked as control variable because it has been used as moderator in some studies. Thus, these demographic variables were controlled in order to enhance the internal validity of this research. The age and employee's tenure were measured in years. The age was coded with 1 for 18–21 years, 2 for 22–30 years, 3 for 31–40 years and 4 for more than 40 years. Employee tenure was measured with 1for less than 3 years, 2 for 3–5 years and 3 for more than 5 years. Education was measured with 1 for high and secondary level, 2 for Master, 3 for MS/M.Phil and 4 for PhD.

## Construct validity of model

For analysis of the data, confirmatory factor analysis (CFA) was used to verify the factor structure and construct validity of the study variables. The three factors (perceived overqualification, creative performance and deviant behaviors) model revealed a good fit $\chi 2$ (167) = 235.0, p\.000, RMSEA = .03, CFI = .97, IFI = .97 to the data. The factors loading for perceived overqualification, deviant behavior and creative performance are .76 to 0.83, 0.68 to 0.84, and 0.62 to 0.78 respectively. All were found significant. Moreover, the differences in chi-square values revealed the best fit for the 3-factors model to the data as compared to different alternative measurement models because the other alternative models have a poor fit to the data (Table 2).

**Table 2. Results of confirmatory factor analyses.**

| Model | $\chi 2$ | df | CFI | IFI | RMSEA |
|---|---|---|---|---|---|
| 1. Three-factor model (PO, DWB & CP) | 235.0 | 167 | .976 | .977 | .038 |
| 2. Two-factor model (PO & DWB merged, CP) | 587.0 | 169 | .787 | .785 | .114 |
| 3. Two-factor model (DWB & CP merged, PO) | 327.0 | 169 | .919 | .919 | .070 |
| 4. Two-factor model (PO & CP merged, DWB) | 342.0 | 169 | .911 | .912 | .073 |
| 5. One-factor model (All items in one factor) | 696.0 | 170 | .729 | .731 | .127 |

PO perceived overqualification, DWB deviant workplace behavior, CP creative performance.

CFI = Comparative Fit Index; RMSEA = Root-Mean-Square Error of Approximation; IFI = Internal Fit Index.

**Table 3. Summary of descriptive statistics.**

| Variables | M | SD | 1 | 2 | 3 | 4 | 5 | 6 | 7 | 8 | 9 |
|---|---|---|---|---|---|---|---|---|---|---|---|
| 1. Gender | 1.66 | .60 | **1.00** | | | | | | | | |
| 2. Age | 0.32 | .46 | .09 | **1.00** | | | | | | | |
| 3. Qualification | 1.90 | .62 | -.02 | .06 | **1.00** | | | | | | |
| 4. Experience | 1.72 | .56 | -.03 | .05 | .55** | **1.00** | | | | | |
| 5. Marital status | 1.84 | .41 | -.11 | .10 | .43** | .37** | **1.00** | | | | |
| 6. Job type | 2.43 | .97 | -.00 | -.04 | -.14** | -.11 | -.05 | **1.00** | | | |
| 7. Overqualification | 2.41 | 1.06 | .00 | -.12 | -.22** | -.16 | -.10 | .03 | (-.92) | | |
| 8. Deviant behavior | 3.67 | .91 | .04 | .08 | .07 | .13 | .07 | -.59** | -.31** | (-.94) | |
| 9. Creative performance | 2.36 | .95 | .06 | -.02 | -.02 | -.06 | -.05 | .30 | .11* | -.20* | (-.76) |

Alpha coefficients appear on the main diagonal.

* $p < .05$.

** $p < .01$.

In sum, in this series of CFAs the results of 3-factors model have revealed that all the variables are discriminately valid. Additionally, it established a foundation for further analysis.

## Descriptive statistics and correlations

Table 3 presents the mean, standard deviation, reliabilities and correlation of the study variables. The values of reliabilities for all variables exceeded the cut off values (.70) as indicated in Table 2. The correlations among variables indicated that overqualification has significant relationship (r = -.31**, p < .01, r = -.11*, $p < .05$) with deviant behavior and creative performance respectively and thus hypotheses 1 and 2 were supported.

## Results

### Testing hypotheses

Results of the regression analysis are presented in Table 4. These results indicate support for hypothesis 1 which states that perceived overqualification has positive association with DWB

**Table 4. Regression analyses examining the moderating effects of personality on stressors- counterproductive work behavior relationships.**

| Criterion variable | Ordered predictors | Job Nature | B | SE | t | P | R² |
|---|---|---|---|---|---|---|---|
| Deviant workplace behaviors | Intercept | | 2.36** [2.25, 2.45] | .05 | 44.5 | 0.01 | |
| | Career stakes | Career | -.45** [-0.59, -0.31] | .03 | -6.34 | 0.01 | |
| | | Survival | .65**[0.637, 0.92] | .06 | 6.62 | 0.01 | |
| | Overqualification (A) | | .38** [0.09, 0.16] | .06 | 6.33 | 0.01 | |
| | A × B | | .27** [0.18, 0.35] | .04 | 6.26 | 0.01 | .12 |
| Creative performance | Intercept | | 2.23** [2.12, 2.33] | .05 | 41.3 | 0.01 | |
| | Career stakes | Career | .63**[0.91, 0.35] | .03 | 4.45 | 0.01 | |
| | | Survival | -.24** [-0.35, -0.11] | .05 | -3.95 | 0.01 | |
| | Overqualification (A) | | .27** [0.39, 0.13] | .06 | 4.15 | 0.01 | |
| | A × B | | -.16** [-0.25, -0.08] | .04 | -3.73 | 0.01 | .06 |

$N = 193$.

* $p < .05$

** $p < .01$.

with beta coefficient ($\beta$ = 0.38, $p$ < 0.001). Similarly, the results also support hypothesis 2 which states that perceived qualification has positive link with creative performance with beta coefficient ($\beta$ = 0.27, $p$ < 0.001).

Table 4 indicates the moderation multiple regression analyses of DWB on the perceived overqualification and career stakes (career or survival job) individually and on their interaction. The moderation of career stakes (career or survival) between the relationship of perceived overqualification and deviant behavior was checked by the interaction affect as shown in Table 4. The interaction effect was found highly significant $\beta$ = 0.27, 95% CI [0.18, 0.35], $t$ = 6.26, $p$ < .001, which indicated that the relationship between perceived overqualification and deviant behavior was moderated by career stakes (career or survival). In addition the value ($\Delta R^2$ = .12) also confirmed the moderation which means that the interaction effect led to an additional 12% variance in DWB, thus H3a and H3b were supported. At high level of career stakes (i.e., career job = 1) the perceived overqualification has a significant negative relationship with deviant behavior $\beta$ = -0.45, 95% CI [-0.59, -0.31], $t$ = -6.34, $p$ < .001. These results confirm the hypothesis H3a in terms of negative interaction effects of career stakes (i.e., career job = 1) in relationship of overqualification and deviant behavior. It means that when an employee perceives job as a career job, she/he will likely experience less deviant behavior. Similarly, at low level of career stakes (i.e., survival job = 0) the perceived overqualification has a significant positive relationship with deviant behavior $\beta$ = 0.65, 95% CI [0.37, 0.92], $t$ = 6.62, $p$ < .001. These results confirm the hypothesis H3b in terms of positive interaction effects of career stakes (i.e., survival job = 0) in relationship of overqualification and deviant behavior. It means that when an employee perceives job as a survival job, she/he will likely experience more deviant behavior. By comparing the values of interaction effects, the value of career stakes (i.e., survival job = 0) as moderator is stronger than the value of career stakes (i.e., survival job = 0).

The moderation of career stakes (career or survival) between the relationship of perceived overqualification and creative performance was checked by the interaction affect as shown in Table 4. The interaction affect was found highly significant $\beta$ = -0.16, 95% CI [-0.25, -0.08], $t$ = -3.73, $p$ < .001, which indicates that the relationship between the perceived overqualification and creative performance was moderated by career stakes (career or survival). In addition, the value ($\Delta R^2$ = .06) also confirmed the moderation which means that the interaction effect leads to an additional 6% variance in creative performance, thus H4a and H4b were also supported. The moderated career stakes (career or survival) was dummy coded with 1 (career job) and 0 for (survival job) in regression analysis. At high level of career stakes (i.e., career job = 1) the perceived overqualification has a significant positive relationship with creative performance $\beta$ = 0.63, 95% CI [0.91, 0.35], $t$ = 4'45, $p$ < .001. These results confirm the hypothesis H4a in terms of positive interaction effects of career stakes (i.e., career job = 1) in relationship of over-qualification and creative performance. It means that when an employee perceives job as a career job, she/he will be prone to demonstrate more creative performance. Similarly, at low level of career stakes (i.e., survival job = 0) the perceived overqualification has a significant negative relationship with creative performance $\beta$ = -0.24, 95% CI [-0.35, -0.11], $t$ = -3.95, $p$ < .001. These results confirm the hypothesis H4b in terms of negative interaction effects of career stakes (i.e., survival job = 0) in relationship of overqualification and deviant behavior. It means that when an employee perceived its job as a survival job will more likely involved in creative performance. By comparing the values of interaction effects, the value of career stakes (i.e., career job = 1) as moderator is stronger than the value of career stakes (i.e., survival job = 0).

The moderating effect of perception of job type (career or survival job) was also tested in Fig 2. through a slope using the process module of Preacher and Hayes [48]. The significant interactions were plotted for high and low (1 below and above SD of the mean) values of the

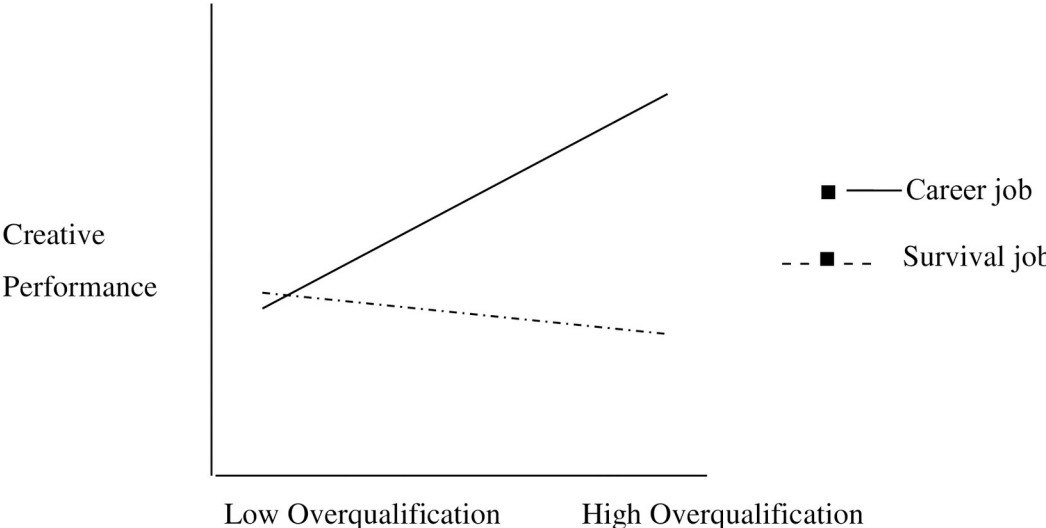

**Fig 2. The interaction effect of perception of job type (career or survival job) and overqualification on creative performance.**

moderators. The graph indicated that when job is perceived as a survival job, the relationship of overqualification and deviant behavior is positively significant and strong enough (i.e., $\beta$ = 0.65,95%CI [0.37, 0.92], $t$ = 6.62, $p < .001$.). While when the employee perceives job as a career job the relationship of overqualification and deviant behavior is negatively significant and is not so much strong (i.e., $\beta$ = -0.45,95% CI [-0.59, -0.31], $t$ = -6.34, $p < .001$). Thus the hypothesis 3 (aggregate of 3a and 3b) was supported.

The moderating effect of perception of job type (career or survival job) was also tested in Fig 3. through a slope using the process module of Preacher and Hayes [48]. The significant interactions were plotted for high and low (1 below and above SD of the mean) values of the moderators. The graph indicated that when job is perceived as a career job, the relationship of

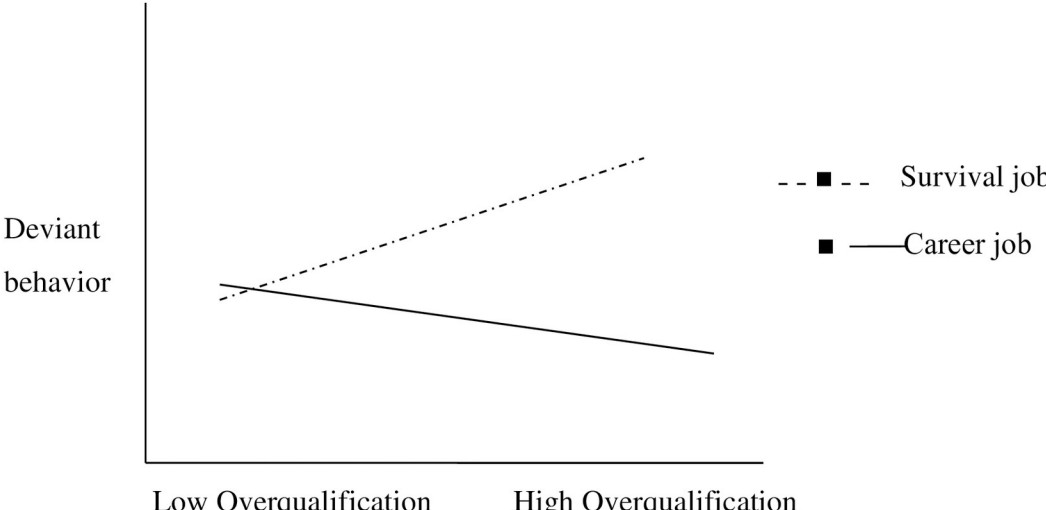

**Fig 3. The interaction effect of perception of job type (career or survival job) and overqualification on deviant behavior.**

overqualification and creative performance is positively significant and strong enough (i.e., $\beta$ = 0.63, 95% CI [0.91, 0.35], $t = 4.45$, $p < .001$). While when the employee perceives job as a survival job the relationship of overqualification and creative performance is negatively significant and is not so much strong (I.e., $\beta$ = -0.24, 95% CI [-0.35, -0.11], $t = -3.95$, $p < .001$). Thus the hypothesis 4 (aggregate of 4a and 4b) was supported.

## Discussion

The extant literature on overqualification carries a negative narrative on overqualified employees. This is because to-date the field has majorly relied on two theories (equity theory and relative deprivation theory) suggesting that overqualification negatively affects performance. However, empirical evidences have been indicating that overqualified employees can demonstrate positively and could be beneficial for the organization. It means that the theoretical explanation of the overqualified employees in the light of equity theory [28] and relative deprivation theory [29] is not sufficient to sustain the negative narrative.

There is a need of critically looking into this narrative. We are required to focus on the potential benefits of overqualification. Keeping this in mind, researchers [16] have emphasized on and have also explored opportunities, conditions and contexts that explain the underlying positive performance consequences of overqualification and their contingencies. There are theories like human capital theory, resources drain theory, and social learning theory which provide explanation that overqualified employees perform better with their surplus KSAs.

The main purpose of this study was to examine the main and interaction effects of perception of job type (career or survival) on the relationship of perceived overqualification to deviant workplace behavior and creative performance. Consistent with the resources drain theory [33, 34] we found a negative significant association between perceived overqualification and deviant behavior. Results exhibited that perceived overqualification is significantly associated with the deviant behaviors. These findings are consistent with the findings of Liu, et al. [1] and Luksyte, et al. [14]. Similarly, results of the study revealed that there is a positive and significant association between perceived overqualification and creative performance. Although in main effects these associations are positive, however, in interaction effects these are either positive or negative.

The main contribution of this study is the introduction of moderator (perceived career or survival) between the perceived overqualification with its tow outcomes deviant behaviour and creative performance. We found that there is strong positive relationship between perceived overqualification and deviant behavior (negative) when employees perceive their current job as a survival job. In contrast, this relationship is negatively strong when job is perceived as career job. These findings are consistent with the notion of resources drain theory which is the theoretical foundation of this relationship. It means when an employee perceives herself/himself overqualified and perceives current job as a survival job ultimately they will not be able to fully exploit the available resources (time and energy) at the current job. Their behavior remains active all the time in search for jobs which fit their qualification well. Therefore, they are more likely perpetrating different types of deviant behaviors (negative). On the other hand, when an employee perceives herself/himself overqualified and perceives current job as a career job are less likely perpetrate deviant acts (negative) but will act positively (even if there is deviation from the norms).

Another important contribution of the study is the measurement of creative performance that is to measure it by customers. According to our knowledge, the extant literature to date has only employed two types of measurements—self measurement and supervisors'

measurement. Several empirical studies have used either self report questionnaire or supervisor report questionnaire with somewhat similar results.

Moreover, we have found that there exists a strong positive relationship between perceived overqualification and creative performance when an employee perceives her/his current job as career job and vice versa. It makes sense that when an employee is overqualified and perceives her/his current job as career job happens to remain more engaged in the current job. Therefore, she/he will utilize much of their resources (time and energy) at the current job and thus will be more likely perform creatively and innovatively. This reasoning best fits with the notion of theory of person-job-fit as well as with the notion of resources drain theory [33].

## Practical implications

The findings of this study have various practical implications. The HR mangers need to be more cautious at the time of recruitment and selection of employees and should only recruit and select those candidates whose overqualification happens to match their career in the current job. It will help employer to mitigate the perpetration of deviant behaviors (negative). Moreover, if employers want to accrue the benefits of high qualification, they should offer them an attractive and secure job where employee can experience the feelings of responsibility, growth, achievement, recognition and, above all, creative performance (positive deviant behaviour). In such scenario the employer can use intrinsic and intellectual capabilities of the overqualified employees for innovation and creativity within workplace. In short, it is the intentions and/or perception of employees about their current job that lead to two completely different and opposite outcomes of the perceived overqualification. However, intentions and perceptions are not created in vacuum. Employees need to be assured of the relevance of their current job with their career. When this relationship is created, overqualification will have positive consequences.

## Limitation and future research

The current study, like most social science research, has some limitations. First, employing of a self-report questionnaire might lead to the problem of common method variance which might affect the results of the study. However some authors have advised that the issue of common method variance is generally overstated [49] so it might not be a serious problem. This issue could be minimized with selection of anonymous sample as we did in this study. Another issue related to the self-report questionnaire is that the respondents were mostly reluctant to respond to the questions of DWB to the extent to which they perpetrate it. Although some studies [e.g., 50, 51, 52] have employed the alternative measures however, their results are same to those studies which have used the self-report questionnaire. It make sense because most of the times employees perpetrate deviant acts without awareness of their supervisor or coworkers [53]. Thus, self-report questionnaire is a valid source to measure the DWB [54]. Second, employing cross sectional design doesn't fully examine the causal relationship because intentions/perceptions might change with the passage of time. Therefore, we suggest the use of longitudinal design in future studies.

This study provides an insight for some in-depth analysis of perceived overqualification-deviant behavior, and perceived overqualification-creative performance relationships. In literature these are mostly studied in the presence of different moderators. Therefore, we suggest opportunities of growth in work organization as a moderator between the relationships of perceived overqualification-deviant behavior that could help to better understand this relationship. Similarly, perceived overqualification-creative performance relationship could be better understood by taking job autonomy as a moderator.

## Conclusion

Close analysis of the extant literature and evidences from empirical studies on overqualification make it evidence that the negative lens with which overqualification has been seen with negative consequences appear myopic. A holistic approach wherein overqualification is seen through positive lens (e.g., its links with creativity and innovation) is more appealing and gives more realistic picture of the reality. This bright side of overqualification with theoretical support would be helpful to inspire researchers to look at the positive aspects and potential benefits of overqualification. We would, therefore, suggest further exploration of various conditions which may help in delving deep into the field.

## Supporting information

**S1 Questionnaire.**
(DOCX)

**S2 Questionnaire.**
(DOCX)

**S1 Data.**
(ZIP)

**S1 File.**
(SAV)

## Author Contributions

**Conceptualization:** Nasib Dar.

**Methodology:** Wali Rahman.

**Supervision:** Wali Rahman.

**Validation:** Wali Rahman.

**Writing – original draft:** Nasib Dar.

**Writing – review & editing:** Wali Rahman.

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
