## [Decision Letter · Decision Letter 0]

17 Oct 2019

PONE-D-19-21532

Deviant behavior and creative performance—the two outcomes of overqualification: The role of career and survival job

PLOS ONE

Dear Dr. Rahman,

Thank you for submitting your manuscript to PLOS ONE. After careful consideration, we feel that it has merit but does not fully meet PLOS ONE’s publication criteria as it currently stands. Therefore, we invite you to submit a revised version of the manuscript that addresses the points raised during the review process.

We would appreciate receiving your revised manuscript by Dec 01 2019 11:59PM. To enhance the reproducibility of your results, we recommend that if applicable you deposit your laboratory protocols in protocols.io, where a protocol can be assigned its own identifier (DOI) such that it can be cited independently in the future. For instructions see: http://journals.plos.org/plosone/s/submission-guidelines#loc-laboratory-protocols

We look forward to receiving your revised manuscript.

Kind regards,

Amelia Manuti

Academic Editor

PLOS ONE

Journal Requirements:

2. Please provide additional details regarding participant consent. In the ethics statement in the Methods and online submission information, please ensure that you have discussed whether participation was voluntary and that the data were collected and analyzed anonymously.

4. Please provide your institutional email address through your Editorial Manager Account.

6. Please ensure that you include a title page within your main document. You should list all authors and all affiliations as per our author instructions and clearly indicate the corresponding author.

Additional Editor Comments:

Dear Authors,

I have now received a blind evaluation of your manuscript from an anonymous reviewer who has considerable expertise in the field. I have also thoroughly read your manuscript. Although a number of strengths were identified, as you will see, the reviews have also highlighted some main shortcomings. Therefore we suggest to substantially revise the manuscript before resubmitting it to the journal. After a close reading of the text, I believe the problems identified by the reviewer and by my self are likely to be successfully addressed even if an hard work is needed.

best regards,

amelia manuti

Reviewers' comments:

Reviewer's Responses to Questions

**Comments to the Author**

1. Is the manuscript technically sound, and do the data support the conclusions?

Reviewer #1: Partly

Reviewer #2: Partly

2. Has the statistical analysis been performed appropriately and rigorously? 

Reviewer #1: Yes

Reviewer #2: Yes

3. Have the authors made all data underlying the findings in their manuscript fully available?

Reviewer #1: No

Reviewer #2: No

4. Is the manuscript presented in an intelligible fashion and written in standard English?

Reviewer #1: No

Reviewer #2: No

5. Review Comments to the Author

Reviewer #1: The manuscript (MS) ‘Deviant behavior and creative performance—the two outcomes of overqualification: The role of career and survival job’ presents a study in a non-WEIRD sample in which hypothesis are tested relating consequences of the overqualification with two criterion variables: deviant behavior and creative performance. The ms subject is relevant mainly because the research was done in substutided culture concerning this and other subjects tipically researched in organizational psychology. The ms has contributions to the literature, but in its present shape it lacks some main issues, which lead me to recommend a revision.

The MS has some central conceptual flaws, such as a clear and explicit definition of deviant behavior and creative performance. The author argues that the use of students as source of information is a contribution to the field of research, but there is no clarity of which measure was applied to students and how it was operationalized. For instance, the paragraph between lines 252-258 that should describe such measure is not clear and produce misunderstandings in the reader when there is mentions of supervisor ratings. Which supervisors?

It is also important to mention that the description of H3 and H4 are inaccurate, considering both hypotheses are impossible to falsify, considering it describe all possible relationships that variables can have. Concerning authors used measures built in different cultures is essential that they providence evidences of cultural adaptation of such measures, considering many different problems that such emerge to this kind of use, such as imposed-ethics.

A minor issue: the MS deserve a careful revision of spelling errors, because there are several throughout the MS.

I consider that the MS has important contributions, but it deserves a careful revision in the above mentioned issues to be suitable to publication in PLOSOne.

Reviewer #2: The paper deals with a very interesting and quite original topic that would be certainly of interest for the readership of the journal: the relationship between overqualification perception and creative performance and deviant behaviors in the workplace. Though an overall positive appreciation I think that the ms has some weaknesses that need to be addressed. First of all, the theoretical section needs to be enriched by a more extensive description of the constructs considered. the authors have articultaed this introductory section into a literature review that carefully considered the ingle relationships between overqualification and creative performance and overqualification and deviant behaviors. however, there is no clear definition neither of deviant behavior nor of creative performance. also career or survaival job is not cleary defined. Further I am not fully conviced about H3 and H4 that sound ambiguous since they consider both positive and negative relationships at the same time. Finally, participants to the study have been both students and teachers. but actually I cannot find reference to how and why students been involved in a study related to overqualification. The conclusion section is too short. I will sugest to make a single section discussion and conclusion and then practical implications and future reserach. Language should also be improved. There are a number of mispelled words. I suggest a native speaker check.

6. PLOS authors have the option to publish the peer review history of their article (what does this mean?). If published, this will include your full peer review and any attached files.

Reviewer #1: No

Reviewer #2: Yes: amelia manuti

---

## [Author Response · Author response to Decision Letter 0]

1 Dec 2019

PLOS ONE Questions: Question No. 1 (Reviewers Response = Partly)

Authors' Response: The authors have gone through the whole paper again and again to address this aspect of the paper. In this effort, additional text has been added.

PLOS ONE Questions: Question No. 2 (Reviewers Response = Yes)

Authors' Response: No explanation is required.

PLOS ONE Questions: Question No. 3 (Reviewers Response = No)

Authors's Response: Data file can be provided when asked for. Authors do not feel any reservations on sharing it with the reviewers and with editors.

PLOS ONE Questions: Question No. 4 (Reviewers Response = No)

Authors' Response: Major portion of the manuscript has been revamped to make it in line with the standard language. Though English is not our first language, we believe that our language is functional if it is not of high standard. We say it with some surety because our whole education is in English and have writing in English since long. 

Comments to the Author (Reviewer # 1)

Comment 1: 

Authors' Response: The contribution of the MS has been appreciated. Therefore, no response is required.

Comment 2:

Authors' Response: To explain the concept of deviant behaviour, a full paragraph (lines 120 to 134) has been added. Efforts have been undertaken to operationalize this concept in the light of overqualification.

Comment 3: 

Authors' Response:Description of the H3 and H4 has been addressed to make it more logically connected and easy to understand.

Comment 4:

Authors' Response: The whole MS has been proofread and a number of concord and other semantic and syntactic issues have been addressed as per understanding of the authors.

Comments to the Author (Reviewer # 2)

Comments

Authors' Response: The authors have tried their level best to address the weakness of the paper as pointed out by the reviewers. Ambiguities in terms of expression in the theoretical section have been addressed. To define the concepts of creative performance, overqualification, and deviant behavior, a full paragraph (lines 120-134) has been added. Similarly, H3 and H4 are aggregate hypotheses. They have sub-hypotheses that H3a and H3b, H4a and H4b respectively. In the figure they are in the same model but actually, they are separately tested and results are discussed. Students have been involved only to have their responses on creativity of the teachers. As they are the direct beneficiaries of their respective teacher creativity, therefore, we took their perception into consideration. In the past, this aspect has generally been assessed through their respective supervisor, we changed supervisors and took students to have more realistic picture. So far conciseness of the conclusion is concerned; we would like to submit that as issues have been discussed in detail in the ‘Discussion’, therefore, conclusion was made terse so as to avoid repetition. The paper has been proofread for spelling and other mistakes and a number of such mistakes have been rectified.

---

## [Editor Report · Decision Letter 1]

5 Dec 2019

Two angles of overqualification-the deviant behavior and creative performance: The role of career and survival job

PONE-D-19-21532R1

Dear Dr. Rahman,

We are pleased to inform you that your manuscript has been judged scientifically suitable for publication and will be formally accepted for publication once it complies with all outstanding technical requirements.

With kind regards,

Amelia Manuti

Academic Editor

PLOS ONE

Additional Editor Comments (optional):

I have had read carefully the authors responses to the two reviews and also the revised manuscript attached. the authors have addressed the main concerns raised and finally the paper is very much improved. In its current form it is a stronger paper that deserves publication. I only suggest a fianl careful reading to amend any grammar error especially within the red sections.
---

## [Editor Report · Acceptance letter]

10 Dec 2019

PONE-D-19-21532R1 

Two angles of overqualification-the deviant behavior and creative performance: The role of career and survival job 

Dear Dr. Rahman:

I am pleased to inform you that your manuscript has been deemed suitable for publication in PLOS ONE. Congratulations! Your manuscript is now with our production department. 

With kind regards,

on behalf of

Dr. Amelia Manuti 

Academic Editor

PLOS ONE